# Illuminating the Unseen: A Large-Scale Exploration of Bias in ICU Discharge Summaries via Language Models

Rui Zhu
*Yale University*
New Haven, CT, USA
rui.zhu.rz399@yale.edu

Genevieve Mortensen
*Indiana University Bloomington*
Bloomignton, IN, USA
gamorten@iu.edu

Yongming Fan
*Purdue University*
West Lafayette, IN, USA
fan322@purdue.edu

Haixu Tang
*Indiana University Bloomington*
Bloomignton, IN, USA
hatang@iu.edu

*Abstract*—Discharge summaries (DS) are pivotal to patient care transitions, yet their completeness and quality can vary widely. This study uses large language models (LLMs) to evaluate DS quality across more than 50,000 ICU discharges from the MIMIC-IV dataset, aiming to quantify compliance with standardized documentation criteria and identify potential biases among different demographic subgroups.

In this study, we adopted 19 established clinical metrics, grouped into five major DS components (e.g., *Reason for Hospitalization*, *Significant Findings*, *Procedures and Treatment*, *Patient's Discharge Condition*, and *Patient and Family Instructions*). Each DS was automatically annotated via LLM-based prompt engineering, producing categorical labels (*Fully, Partial, Unacceptance, Missing*). We then conducted numeric score-based and level-wise statistical analyses to detect variations in DS quality across race, insurance type, chief complaints, and admission types.

For the results, while *Reason for Hospitalization* was generally well documented, up to 10% of DSs lacked sufficient *Patient and Family Instructions* and 3–10% had incomplete *Discharge Condition* details. Statistically significant disparities ($p < 0.05$) were observed among subgroups, with higher negative scores (i.e., *Missing* or *Unacceptable*) in certain demographic categories, notably Asian males insured under less common plans (not in Medicare or Medicaid), where over 7% of DSs contained deficiencies—more than twice the overall average.

*Index Terms*—Large Language Model, Discharge Summary, Bias

## I. INTRODUCTION

Discharge summaries (DS) are critical documents that encapsulate a patient's hospital course, diagnoses, treatments, and follow-up instructions. They serve as an essential communication tool between healthcare providers, ensuring continuity of care and reducing the risk of medical errors as patients transition between care settings [1], [2]. In the Intensive Care Unit (ICU), this role becomes even more pronounced due to the complexity and acuity of patient conditions. ICU patients often undergo multiple procedures, experience rapid physiological changes, and require intricate medication regimens [3]. Consequently, accurately captured discharge information is vital for downstream providers—ranging from step-down units to subacute facilities—so they can anticipate patients' needs and align care plans accordingly.

In the United States, the Joint Commission (JC) is a widely recognized, independent, non-profit organization that accredits and certifies healthcare institutions [4]. It sets performance standards aimed at improving healthcare quality and patient safety. Among these standards, *Standard IM.6.10, EP 7* outlines six mandated components for discharge summaries, which also inform best practices in the ICU setting due to their emphasis on clear communication at a high-risk juncture. These components include: **(1) Reason for hospitalization. (2) Significant findings. (3) Procedures and treatment provided. (4) Patient's discharge condition. (5) Patient and family instructions. (6) Attending physician's signature.**

Prior work has investigated the extent to which hospital discharge summaries meet these Joint Commission guidelines. For example, Smith *et al.* [5] analyzed 599 discharge summaries from an academic hospital in the Midwestern United States (2003–2005). Their results showed high overall compliance (88–100%) with five of the six mandated elements. However, "patient's discharge condition" was documented least frequently (79–90%), particularly among stroke patients, potentially compromising post-discharge care planning [6]. Despite these valuable findings, many such studies rely on expert human reviewers, who manually rate DS completeness in relatively small samples. While this approach is precise, it limits scalability and timeliness. Furthermore, large-scale evaluations of ICU discharge summaries—where clinical complexity is greater and documentation demands are higher—have remained understudied.

In parallel, recent developments in *large language models* (LLMs) [7]–[11] offer new opportunities to systematically assess clinical notes, including discharge summaries, on a more expansive scale [12]–[14]. LLMs are trained on vast corpora of text and can exhibit remarkable comprehension of clinical language, enabling them to extract, summarize, and classify information with minimal human intervention. These models have demonstrated promise in tasks ranging from section labeling in electronic health records to automatic generation of coherent summaries for patient handoffs [14], [15]. By leveraging LLMs, it becomes feasible to evaluate thousands of ICU discharge summaries rapidly, thereby detecting gaps and

potential biases that might otherwise go unnoticed in smaller, manually reviewed datasets.

In this study, we harness the capabilities of LLMs to examine a large-scale dataset of ICU discharge summaries and assign quality labels based on established clinical documentation criteria. By applying this automated approach, we aim to:

- Assess how well modern ICU discharge summaries conform to the Joint Commission's recommended components;
- Investigate potential biases in DS quality across different demographic groups and clinical conditions;
- Uncover previously unrecognized patterns or deficiencies that may inform interventions for improved continuity of care.

Our analysis of more than 50,000 ICU discharge summaries reveals that although certain core elements (such as *Reason for Hospitalization*) tend to be well documented, other critical components—particularly *Patient and Family Instructions* and *Patient's Discharge Condition*—are often missing or insufficiently recorded, with up to 10% of summaries exhibiting such gaps. In addition, we observe marked disparities among specific demographic subgroups; for instance, summaries prepared for Asian male patients who rely on insurance coverage *not classified under standard government or private plans* have a rate of missing or unacceptable information exceeding 7%, which is more than twice the 3.5% average in our overall sample. Statistical testing (e.g., $p < 0.05$) confirms the significance of these differences.

## II. METHODS

### A. Metrics Design and Annotation Scheme

Following the guidelines from a previous study on Discharge Summary (DS) quality [5], we adopted 19 distinct metrics grouped into five major categories:

A **Reason for Hospitalization**: Refers to the chief complaint and/or the history of present illness.

B **Significant Findings**: Refers to the primary diagnoses.

C **Procedures and Treatment Provided**: Includes the hospital course and/or any hospital consults or procedures.

D **Patient's Discharge Condition**: Captures any documentation conveying the patient's status at discharge.

E **Patient and Family Instructions**: Includes instructions provided to the patient and/or family members, when appropriate.

Originally, a sixth category named *Signature* was part of the official guidelines, but it was omitted here because MIMIC-IV deidentification removes provider names and signatures.

For each of the five main categories (A–E), we used up to four evaluation angles:

1 **Prioritized:** Information is displayed in a coherent order of importance, with signs, symptoms, tests, and procedures properly organized (including the care plan).

2 **Sufficient:** The summary provides enough relevant details for its clinical purpose.

3 **Clear:** The information is easily understood by healthcare providers or other relevant readers.

4 **Concise:** The information is focused, succinct, and largely free from redundancy.

Because category D (Patient's Discharge Condition) typically contains only a brief statement (e.g., "patient stable at discharge"), we found that *Prioritized* was not particularly relevant there, so D only uses the remaining three angles. Overall, this yields 19 sub-items (A1–A4, B1–B4, C1–C4, D1–D3, E1–E4).

For each discharge summary, we prompted a large language model (LLM) to assign one of four labels: *Fully, Partial, Unacceptance, or Missing* to each sub-item. These labels reflect a gradient of completeness, from having all necessary details (*Fully*) to entirely lacking the targeted information (*Missing*). We did not involve human annotators in this process, since prior studies have demonstrated high concordance between LLM-based annotation and expert judgment for ICU discharge summaries. In subsequent analyses, we refer to each sub-item by its letter-number notation (e.g., A1 for the first question in category A, B3 for the third question in category B).

### B. Prompt Engineering Strategy

Although large language models (LLMs) have demonstrated impressive capabilities in parsing and summarizing lengthy clinical texts, practical limitations on model context windows and memory retention pose challenges when multiple questions must be answered sequentially. In our case, each ICU discharge summary (DS) could reach approximately 2,000 words, and we needed the LLM to evaluate 19 specific items organized under five major aspects of discharge documentation. We discovered that bundling all 19 items into a single query often led to incomplete or inconsistent answers, presumably because the model struggled to "remember" all components as the prompt expanded in length.

To address this, we designed a multi-step prompt approach: for each DS, we engaged in *five separate mini-sessions*, each corresponding to one of the five key aspects (e.g., "Reason for Hospitalization," "Significant Findings," etc.). Within each mini-session, the LLM was asked to evaluate only the subset of items tied to that aspect. This structure allowed the model to focus its context window on a narrower set of instructions and excerpted text, thereby reducing cognitive load and improving answer consistency.

*a) Four-Option Constrained Response:* Another challenge was ensuring that the LLM's outputs were both machine-readable and clinically interpretable. While free-text responses can be rich and nuanced, they also complicate downstream analysis. Therefore, we adopted a strict four-option format for each item: Fully, Partial, Unacceptance, or Missing.

We instructed the model to provide only these labels (and no other commentary) for each DS item, effectively *forcing the response into a categorical output*. This approach had two benefits: (1) it simplified subsequent data processing and statistical analyses, and (2) it minimized the risk of irrelevant or off-topic generation by the LLM.

*b) Role and Task Specification:* To further guide the model, we explicitly specified its role and the nature of the task. We found that *role-based instructions* led to more contextual and succinct responses, matching the style of a clinical documentation auditor or attending physician. Although we tuned exact phrasing through iterative experimentation, the final version of our prompt resembled the following structure:

> **Role:** You are the attending physician on record, auditing an ICU discharge summary to ensure complete documentation.
> **Task:** Assess the summary for five main aspects, with sub-items under each aspect. Assign one of the following labels to each sub-item: *Fully, Partial, Unacceptance, or Missing.*
>
> **Selected Aspect (Example):** Reason for Hospitalization
> **Definition:** Chief complaint and/or history of present illness.
> **Items to Evaluate:**
> Q1. Sufficient information (enough information for its purpose; includes pertinent details)
> Q2. Concise (focused, brief, not redundant)
> Q3. Clear (understandable to providers and others)
> Q4. Organized (properly grouped, chronological, can find important information)
>
> **Discharge Summary Excerpt:** [*Full DS text inserted here*]
>
> **Output Format:** Provide four labels corresponding to items 1 through 4, separated by commas (e.g., "Fully, Partial, Unacceptance, Missing").

By providing a consistent structure—defining the role, clarifying the items to be evaluated, and constraining the output to a small set of specific labels—we significantly improved the reliability and clarity of the responses. Each of the five mini-sessions followed a similar format, covering all 19 items in multiple passes rather than a single conversation.

*c) Overcoming Context and Consistency Challenges:* Initial testing revealed that if we combined all 19 items into a single prompt, the model sometimes produced incomplete label sets or drifted into free-text responses. Splitting the 19 items into five targeted queries, along with repeated reminders of the expected response format, curbed these issues.

To eliminate stochastic variation, we set the inference temperature to 0 for every run, which guarantees deterministic outputs for identical inputs. We also constrained the model to return only the predefined labels; this design greatly reduces the model's sensitivity to minor wording changes [16]–[18]. In practice, we experimented with several semantically equivalent rephrasings of the prompts (e.g., swapping clause order or synonyms) and observed no change in the assigned labels; an observation consistent with prior work showing that GPT-4 is stable under small prompt perturbations in structured classification tasks [19], [20].

Moreover, we found that the introduction of role-specific instructions (e.g., "attending physician," "clinical documentation auditor") led to more clinically oriented language, suggesting that the model was better aligned with the domain context.

Overall, this prompt engineering strategy allowed us to handle lengthy ICU discharge summaries while still capturing structured feedback across a wide range of quality metrics. Our final system thus balances the depth of textual analysis (via an LLM) with the practical need for clean, standardized outputs amenable to large-scale statistical evaluation.

*C. Bias Detection and Analysis Methods*

We investigated whether certain demographic subgroups (e.g., race, insurance type, marital status) receive systematically higher or lower metric assessments. Recognizing that direct numeric encoding of qualitative categories can be convenient yet potentially arbitrary, we conducted two complementary analyses:

*a) Numeric Score-Based Analysis:* For a quantitative summary of each discharge summary, we mapped the four LLM-generated labels to integer scores:

$$\text{Fully} = 3, \quad \text{Partial} = 2, \quad \text{Unacceptance} = 1, \quad \text{Missing} = 0.$$

Let $X_{ij}$ denote the encoded score of metric $i$ for the $j$-th discharge summary. We then evaluated whether these numeric scores differed significantly across subgroups. If $X_{ij}$ satisfied approximate normality assumptions, we applied a one-way analysis of variance (ANOVA). The test statistic $F$ is computed as:

$$F = \frac{\text{Mean Square Between Groups}}{\text{Mean Square Within Groups}}$$

where the *mean square between groups* is the sum of squared deviations of each group mean from the overall mean, divided by the number of groups minus one; and the *mean square within groups* is the sum of squared deviations of individual observations from their respective group means, divided by the total number of observations minus the number of groups.

*b) Level-Wise Distribution Analysis:* To retain the categorical integrity of the four-level annotation (*Fully, Partial, Unacceptance, Missing*), we performed a chi-square test of independence to determine whether the proportions of each label differed significantly across demographic subgroups. Let

$$\mathbf{O} = \{O_{c,s}\} \quad \text{and} \quad \mathbf{R} = \{R_{c,s}\}$$

represent the observed counts (e.g., for a Latino population) and corresponding reference counts (e.g., for a White population), respectively, where

$$c \in \{\text{Fully}, \text{Partial}, \text{Unacceptance}, \text{Missing}\}$$
$$\wedge \ s \in \{\text{Subgroup}_1, \ldots, \text{Subgroup}_k\}$$

The chi-square statistic is computed as:

$$\chi^2 = \sum_c \sum_s \frac{(O_{c,s} - R_{c,s})^2}{R_{c,s}}$$

which quantifies the degree to which the observed category counts deviate from those in the reference group. A significance level of $\alpha = 0.05$ was used to determine whether any subgroup's distribution of labels diverged from the reference distribution.

*c) Multiple-Comparison Adjustment:* Because both the numeric and level-wise analyses involve several related hypothesis tests, reporting raw $p$-values alone would inflate the family-wise Type-I error. To keep the false-discovery rate (FDR) under control, we therefore applied the Benjamini–Hochberg procedure (BH) [21] with a target $q = 0.05$ to every logical family of comparisons—race (4 tests), insurance (2 tests), chief complaint (4 tests), and admission type (3 tests).

## III. RESULTS

### A. Data Source and Preprocessing

We used deidentified discharge summaries from the MIMIC-IV Intensive Care Unit (ICU) dataset [22], yielding a total of 50,496 documents. Each record captures a narrative of the patient's clinical course during their ICU stay, relevant interventions, and instructions at discharge. All discharge summaries labeled as "ICU Discharge Summaries" were included without additional exclusion. Table I summarizes the distribution of patient demographics (e.g., race, insurance type, marital). All patient data in MIMIC-IV are deidentified, and our use of the dataset complies with its guidelines requirements.

Table I summarizes the MIMIC-IV ICU discharge summaries used in this study. We consolidated the original MIMIC race labels into five broader categories—White, African, Hispanic, Native, and Asian—to facilitate clearer bias analyses. Specifically, the *White* group encompassed all entries labeled "WHITE," "WHITE - OTHER EUROPEAN," "WHITE - RUSSIAN," "PORTUGUESE," or other European-origin designations. The *African* group comprised entries such as "BLACK/AFRICAN AMERICAN," "BLACK-/CAPE VERDEAN," "BLACK/CARIBBEAN ISLAND," and "BLACK/AFRICAN." The *Asian* category included "ASIAN," "ASIAN - CHINESE," "ASIAN - SOUTH EAST ASIAN," "ASIAN - ASIAN INDIAN," and similar labels. The *Native* group was formed from any entries referencing "NATIVE" or "AMERICAN," while the *Hispanic* group contained those labeled "HISPANIC," "LATINO," or "SOUTH AMERICAN." All other or ambiguous designations were grouped as unknown, and we do not consider in this study.

We also consolidated the numerous chief complaints documented in MIMIC into five main categories (Cardio, Pulmonary, Neuro, GI, and Other), guided by the principal physiological system or clinical context of each complaint. For instance, presentations involving cardiac or vascular issues were classified under Cardio, respiratory symptoms under Pulmonary, neurological conditions under Neuro, and digestive system problems under GI, while any remaining or multi-faceted complaints were placed in Other. This standardized approach to both race and chief complaint classification allowed for a more tractable and interpretable stratification of the patient population in subsequent analyses.

We classified the MIMIC admission records into five categories based on their original admission types in the database. Emergent admissions comprise entries labeled "EW

EMER." and "DIRECT EMER." in MIMIC, signifying an emergency or direct emergency pathway (often from the Emergency Department). Observation admissions, including "OBSERVATION ADMIT," "EU OBSERVATION," "DIRECT OBSERVATION," and "AMBULATORY OBSERVATION," represent short-term stays intended for monitoring and assessment rather than full inpatient care. Urgent admissions derive from the "URGENT" label, indicating a high-acuity but non-elective need for hospitalization. Elective admissions, taken from "ELECTIVE" and "SURGICAL SAME DAY ADMISSION," are planned or scheduled hospital entries, commonly for routine surgeries or procedures. Finally, any admission types not fitting the above criteria are grouped into Other, ensuring comprehensive coverage of all MIMIC admission designations.

**Large Language Models Used in this Study**

We leveraging the `GPT4_0613` model via the Azure OpenAI service. Compared to previous iterations (e.g., GPT-3.5), GPT-4 demonstrates marked improvements in linguistic coherence, reasoning ability, and factual reliability [23], enabling more precise extraction and categorization of relevant clinical details. Early validations in medical settings have shown that GPT-4 can match or exceed human-level performance on tasks such as summarizing patient records and identifying salient diagnoses [24]. This increased fidelity and domain adaptability make GPT-4 especially suited to the high-stakes environment of ICU documentation, where granular accuracy in discharge summaries is paramount for safe and effective care transitions.

### B. Overall Quality of Discharge Summaries

Figure 1 provides an overview of the score distribution for all 19 discharge summary metrics (labeled A1 through E4). The horizontal axis lists each metric, while the vertical axis indicates the proportion of four possible ratings (*Missing, Unacceptable, Partially, Fully*). For clarity, positive ratings (*Fully* and *Partially*) appear above a central baseline, whereas negative ratings (*Missing* and *Unacceptable*) appear below it.

Several notable patterns emerge. First, metrics in the "E" category, which relate to patient and family instructions, exhibit a relatively higher proportion of incomplete documentation: approximately 10% of discharge summaries are labeled *Missing* for these items. Additionally, "D" category metrics (Patient's Discharge Condition) include 3–10% of summaries with *Unacceptable* or *Missing* ratings, indicating that many DSs provide suboptimal information regarding patients' status upon discharge. In contrast, the "A" category (Reason for Hospitalization) receives fewer negative scores and boasts 30–50% of DSs classified as *Fully*, suggesting that the rationale for admission is typically well documented. Overall, these results imply that while most core elements are addressed adequately, there remain specific areas—particularly instructions to patients and families, as well as discharge condition details—where additional focus is warranted.

| Dimension | Categories (Count) | | | | |
|---|---|---|---|---|---|
| **Race** | White (34,100) | African (799) | Hispanic (1,883) | Native (4,075) | Asian (1,485) |
| **Insurance** | Other (25,075) | Medicare (21,828) | Medicaid (3,593) | | |
| **Chief Complaints** | Cardio (5,104) | Pulmonary (7,621) | Neuro (7,225) | GI (4,741) | Other (25,805) |
| **Admission Type** | Emergent (24,202) | Observation (10,858) | Urgent (8,358) | Elective (7,078) | |

TABLE I: Distribution of MIMIC-IV ICU Discharge Summaries

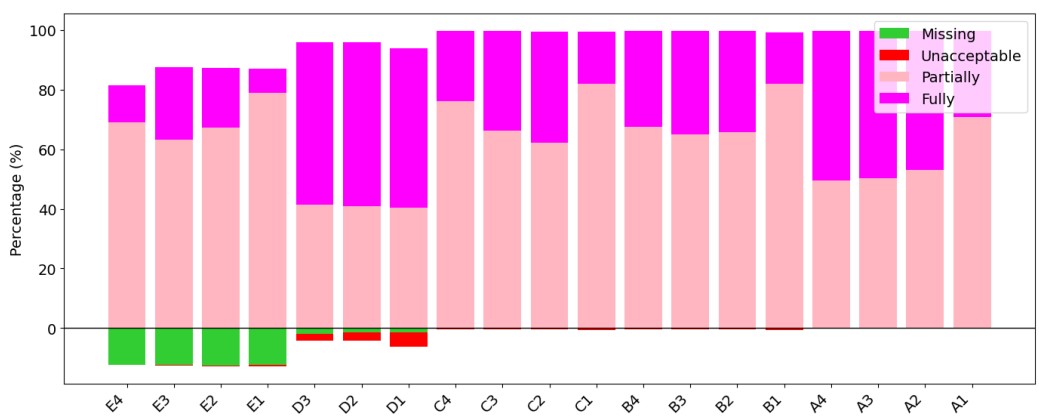

Fig. 1: Overall Quality Distribution in 19 Different Metrics

### C. Population-Based DS Quality and Bias Analysis

We analyzed discharge summary quality across multiple demographic subgroups, focusing on sex, race, and insurance type. To quantify potential biases, we used both numerical score-based methods and level-wise distribution analyses (see Table II), examining whether certain subgroups consistently exhibited higher frequencies of missing or unacceptable documentation. To illustrate overall trends, we first merged each DS's 19 metric labels into a single composite distribution representing *negative outcomes* (i.e., "Missing" or "Unacceptable"). Figure 2a compares three insurance categories along this dimension, showing that DSs associated with "Other" or "Medicaid" insurance have similar rates of *Missing* (roughly 2.5%) and *Unacceptable* (around 0.5%). In contrast, the "Medicare" subgroup exhibits a higher *Missing* rate of approximately 3.5% and a slightly larger *Unacceptable* fraction. Statistical tests confirmed that these differences are significant ($p < 0.05$) under both our numeric scoring scheme and the level-wise chi-square analysis.

Figure 2b displays analogous results by race, indicating that the "Asian" subgroup has the highest overall proportion of negative DS outcomes (around 4%), while the lowest is observed for the "African" subgroup (approximately 2.5%). Again, our combined tests suggest these differences cannot be solely attributed to chance. To further explore intersectional effects, we conducted a fine-grained investigation of race, sex, and insurance status. Figure 2c highlights an especially vulnerable cohort—Asian males with "Other" insurance—who collectively produced 830 discharge summaries. In the MIMIC dataset, "Other" typically denotes coverage not classified under standard government or private plans, encompassing a variety of less common insurance arrangements. Over 7%

of these discharge summaries had *Missing* or *Unacceptable* ratings, more than double the 3.5% average observed across the broader dataset. Both numeric and categorical analyses corroborated the statistical significance of this disparity ($p < 0.05$).

These findings underscore the importance of monitoring DS quality across diverse populations, as certain subgroups may require targeted interventions to enhance completeness and clarity in their discharge documentation.

Table II displays the results of both numeric score-based and level-wise distribution analyses for comparing DS quality across race and insurance subgroups, using White (for race) and Other (for insurance) as reference categories. Here, each DS's 19 item scores were merged into a single composite rating for these difference tests. Both numeric and level-wise $p$-values are presented, all of which are below the 0.05 threshold, indicating statistically significant group differences.

### D. Chief Complaint and Admission-Based DS Quality and Bias Analysis

In a parallel fashion to our population-based evaluation, we also examined DS quality differences by patients' Chief Complaint (CC) and Admission Type. First, Figures 3 and 4 present the score distributions for each subgroup, using the same merged metric categories (A, B, C, D, and E) as before. Next, Table III summarizes the statistical significance of these observed differences under both the numeric score-based and level-wise distribution tests.

Specifically, Figure 3 illustrates how each Chief Complaint group (e.g., Cardio, Pulmonary) fares across the five merged metric categories. On the horizontal axis, we plot the proportion of *positive* (Fully or Partially) versus *negative* (Missing or Unacceptable) scores, while the vertical axis aligns the metrics

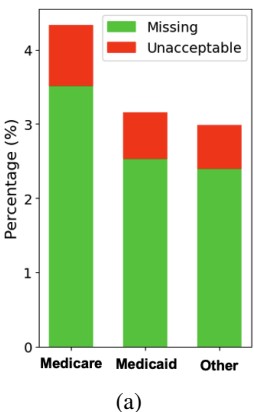
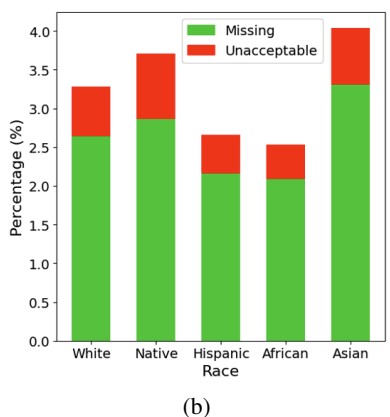
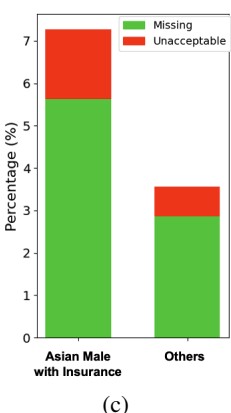

(a)             (b)             (c)

Fig. 2: (**a**) Negative outcome distributions (i.e., *Missing* or *Unacceptable*) for three insurance types, (**b**) major racial subgroups, and (**c**) an intersectional subgroup of Asian males with Non-medicare and non-Medicaid.

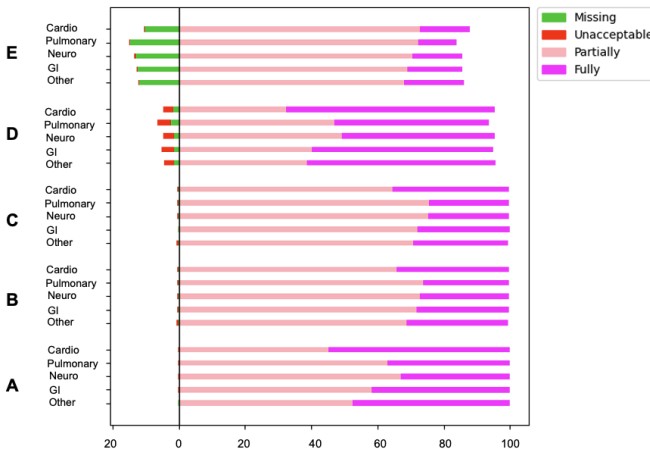

Fig. 3: Diverging Bar Chart for Chief Complaint          Fig. 4: Diverging Bar Chart for Admission Type

by Condition. From these data, *Cardio* generally achieves higher positive ratings and fewer negative ones, suggesting better overall DS quality than other groups. By contrast, *Pulmonary* shows less favorable outcomes, particularly for the D and E metrics, where its negative score proportions exceed those of other subgroups. Even for metrics A, B, and C (which typically record fewer negative ratings), Pulmonary exhibits a notably smaller fraction of *Fully* ratings, lagging behind the leading subgroup by about 4–19%.

Figure 4 displays a similar analysis for Admission Type. Among the four types examined (*Emergent, Observation, Urgent, Elective*), the *Elective* group consistently achieves better performance across all five metric categories. It has the lowest prevalence of negative ratings and the highest proportion of *Fully* or otherwise positive scores. Meanwhile, the remaining three Admission Types show minimal differences in four of the five metric categories. One exception is category E, where *Observation* fares slightly better than both *Emergent* and *Urgent*. Overall, these findings underscore potential variations in DS completeness and clarity driven by both patients' clinical conditions and the nature of their hospital admissions.

We further quantified these differences in Table III, conducting both numeric-based and level-wise statistical tests for each subgroup. For Chief Complaint (CC), we used the largest cohort, *Other*, as the reference and compared each specific CC group against it. All resulting $p$-values were under 5%, indicating significant differences relative to the *Other* category. For Admission Type, we similarly used the largest cohort, *Emergent*, as the reference. Apart from the comparison between *Urgent* and *Emergent*, which yielded a $p$-value above 5%, all other comparisons showed $p$-values below the 5% threshold. Hence, *Urgent* and *Emergent* exhibit no statistically significant difference, whereas the remaining Admission Types diverge significantly from *Emergent*.

## IV. DISCUSSION

Ensuring the integrity of discharge summaries in the ICU setting is crucial for patient safety and care continuity. Our study reveals that while certain elements of DSs (e.g., *Reason for Hospitalization*) are adequately covered, other components—particularly *Patient and Family Instructions* and *Discharge Condition*—continue to exhibit notable gaps. These

| Race vs. White | | |
| --- | --- | --- |
| **Race** | **Numeric $p$ / BH $q$** | **Level-wise $p$ / BH $q$** |
| African | 0.030 / 0.040 | 0.028 / 0.042 |
| Hispanic | 0.017 / 0.034 | 0.042 / 0.042 |
| Native | 0.045 / 0.045 | 0.038 / 0.042 |
| Asian | 0.008 / 0.032 | 0.006 / 0.024 |
| Insurance vs. Other | | |
| **Insurance** | **Numeric $p$ / BH $q$** | **Level-wise $p$ / BH $q$** |
| Medicare | 0.022 / 0.022 | 0.027 / 0.034 |
| Medicaid | 0.018 / 0.022 | 0.034 / 0.034 |

TABLE II: Differences in DS Quality by Race and Insurance.

| Chief Complaint vs. Other | | |
| --- | --- | --- |
| **CC** | **Numeric $p$ / BH $q$** | **Level-wise $p$ / BH $q$** |
| Cardio | 0.024 / 0.032 | 0.026 / 0.041 |
| Pulmonary | 0.045 / 0.045 | 0.047 / 0.047 |
| Neuro | 0.012 / 0.032 | 0.008 / 0.032 |
| GI | 0.019 / 0.032 | 0.031 / 0.041 |
| Admission Type vs. Emergent | | |
| **Type** | **Numeric $p$ / BH $q$** | **Level-wise $p$ / BH $q$** |
| Observation | 0.038 / 0.057 | 0.029 / 0.044 |
| Urgent | 0.244 / 0.244 | 0.239 / 0.239 |
| Elective | 0.021 / 0.057 | 0.025 / 0.044 |

TABLE III: Differences in DS Quality by Chief Complaint and Admission Type.

omissions can hinder effective transitions between acute and post-acute care, resulting in suboptimal treatment planning, delayed rehabilitation, and potential readmissions.

The observed subgroup disparities, especially those affecting Asian males with alternative insurance coverage, highlight potential systemic biases in how discharge documentation is approached. These biases might stem from institutional workflows, the nature of payer-provider relationships, or even language and cultural barriers affecting patients. Similarly, the variability in DS quality across different admission types and chief complaints underscores the importance of context-specific documentation protocols: conditions requiring more specialized or emergent interventions may be at higher risk for omissions in discharge documentation.

Our approach harnessing LLMs underscores the potential of large-scale automated tools to tackle the labor-intensive process of DS auditing. Although the model generally aligned with expert assessments in prior work, we acknowledge that LLMs—trained primarily on publicly available corpora—may have limitations in comprehensively understanding unique medical terminologies or rare clinical nuances. Further, prompt engineering strategies remain subject to continuous refinement, and model outputs can be influenced by the design of specific instructions.

Because annotating more than fifty-thousand ICU discharge summaries by hand is impractical, we relied on GPT-4 for all labels, a choice that inevitably risks mis-handling rare terminology and silently propagating model bias. To quantify

that risk, our next phase research will draw a stratified sub-sample for blinded dual review by clinical experts and will report inter-rater $\kappa$ versus the LLM. Misclassified cases will seed an active-learning loop that refines prompts and post-processing rules. We will also replicate the pipeline at two external hospitals to test whether both model performance and the disparity patterns observed here generalise beyond the single-centre MIMIC-IV context.

Additionally, while our statistical findings suggest meaningful associations, the analyses do not fully disentangle causal factors. Real-world clinical processes are multi-faceted, and documentation lapses can be influenced by staff work-load, training, EHR interface design, or individual clinician preferences. Understanding these contextual elements would likely benefit from mixed-methods approaches, incorporating qualitative interviews or focus groups alongside quantitative data analysis.

The present study stops at describing documentation gaps; it does not yet ask whether those gaps translate into harm. As a logical next step, we will merge our quality labels with longitudinal outcome data, such as 30-day readmissions, emergency visits, medication errors, and mortality, available in follow-up registries. Multivariable Cox and logistic models will be employed to test whether summaries flagged as "Missing" or "Unacceptable" in key sections predict higher event rates after case-mix adjustment. Confirming such links would turn our descriptive findings into actionable quality-improvement targets and provide an empirical foundation for real-time LLM interventions at the point of discharge.

Looking ahead, our work demonstrates that automated large-scale DS evaluation can pinpoint under-documented elements and highlight potential demographic disparities. These findings can guide targeted interventions, such as tailored education for care teams, standardization of discharge check-lists, or improved EHR prompts. Moreover, deploying such automated evaluations in near-real-time might enable timely correction of DS deficiencies before patient transfer, thereby optimizing care continuity.

## V. CONCLUSION

This study leverages large language models to systematically assess discharge summary quality across a vast corpus of ICU records, revealing both strengths—such as well-documented reasons for hospitalization—and significant deficiencies, particularly in *Patient and Family Instructions* and *Discharge Condition*. The presence of clinically meaningful disparities among certain demographic subgroups suggests an ongoing need for targeted corrective measures to ensure equitable care transitions. Overall, our findings highlight the potential of automated NLP-based approaches to streamline documentation review and enhance patient safety. Future investigations should examine how these methods can be integrated into clinical workflows, whether real-time feedback can be used to improve DS quality, and how better discharge documentation correlates with patient outcomes across diverse healthcare settings.

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
