# OpenReview forum: "Illuminating the Unseen: A Large-Scale Exploration of Bias in ICU Discharge Summaries via Language Models"
_IEEE.org/EMBS/BHI/2025/Conference — BHI 2025 Oral_

### Official Review · Reviewer_yuNj · 2025-06-26
**Innovative method to highlight biases in clinical documentation**

**Confidence:** 5
**Clarity Of Writing:** excellent
**Clinical Significance:** excellent
**Methodological Novelty:** great
**Overall Rating:** 8

**Experiments And Results:**

excellent

**Questions For The Authors:**

My only question would be how do you see this LLM-based system being integrated into a hospital's workflow to prevent these documentation gaps before a patient is discharged, rather than just identifying them later?

**Strengths:**

The use of LLMs to analyze a massive dataset of over 50,000 ICU discharge summaries overcomes the limitations of previous studies that relied on smaller samples and labor-intensive manual reviews by human experts. This supports a more comprehensive evaluation of documentation quality. The study uses a detailed framework with 19 distinct metrics organized into five clinical components, such as "Reason for Hospitalization" and "Patient and Family Instructions". Each metric was assessed using 4 evaluation angles: Prioritized, Sufficient, Clear, and Concise. This method provides an assessment of discharge summary quality beyond simple completeness checks. The paper identifies disparities in the quality of discharge summaries across various subgroups, suggesting that certain populations, such as Asian males with non-standard insurance, may be more likely to receive documentation with deficiencies at rates exceeding twice the average.

**Summary Of The Paper:**

Large language models (LLMs) were applied to analyze over 50,000 ICU discharge summaries from the MIMIC-IV dataset. The goal was to assess how well these summaries aligned with standard documentation practices and to explore potential disparities across demographic groups.
The study used 19 clinical metrics grouped into five categories: Reason for Hospitalization, Significant Findings, Procedures and Treatment, Patient's Discharge Condition, and Patient and Family Instructions. GPT-4 was used to annotate each summary, assigning one of 4 categorical labels, "Fully," "Partial," "Unacceptable," or "Missing", to each metric. To assess bias, the researchers translated these labels into numerical scores (ranging from 0 to 3) and performed chi-square tests to evaluate differences across groups. Race labels were consolidated into 5 larger categories, and both chief complaints and admission types were grouped into more general classifications.
Several documentation gaps emerged from the analysis. While "Reason for Hospitalization" was typically well-recorded, up to 10% of summaries lacked adequate "Patient and Family Instructions," and 3-10% were missing details about the patient's discharge condition.
The findings also pointed to disparities tied to demographic factors with statistically significant differences (p < 0.05) were observed across subgroups. As an example, discharge summaries for Asian male patients with less common types of insurance had over 7% missing or poor-quality information. Within group comparisons showed that summaries for Asian patients had the highest proportion of negative documentation outcomes (around 4%), while the African subgroup had the lowest (approximately 2.5%).
Discharge summaries for patients with Medicare were more likely to contain missing information (about 3.5%) compared to those with Medicaid or other insurance types (approximately 2.5%). Documentation quality also varied by chief complaint and admission type, such as summaries linked to cardiac-related complaints were more complete, whereas those associated with pulmonary conditions were less so. Elective admissions tended to have better documentation quality than emergent, observation, or urgent admissions.
In general, the study suggests that while many components of ICU discharge summaries are well-covered, important gaps remain, particularly regarding discharge condition and patient instructions. These gaps, along with the disparities identified across demographic and clinical subgroups, may reflect broader systemic issues. The authors propose that large-scale, automated evaluations using LLMs could support efforts to monitor and improve documentation quality, ultimately contributing to more equity across care.

**Weaknesses:**

The study's methodology entirely depends on an LLM for data annotation without appearing to incorporate human validation for this specific task. The authors justify this by citing high LLM-human concordance in prior studies, but they also acknowledge that LLMs may have "limitations in comprehensively understanding unique medical terminologies or rare clinical nuances". The authors clearly state that while their statistical analysis reveals significant associations between demographic groups and documentation quality, it "do[es] not fully disentangle causal factors". The observed disparities could be influenced by some unmeasured variables, such as "staff workload, training, EHR interface design, or individual clinician preferences".  The findings are derived from a single data source, the MIMIC-IV dataset, which contains records from one academic medical center in the United States. This reliance on a single source limits the generalizability of the findings. Factors like institutional workflows, payer-provider relationships, and regional patient populations can vary between healthcare settings, meaning the biases and documentation gaps identified may not be representative of other hospitals or healthcare systems.

---

### Official Review · Reviewer_sTEb · 2025-07-15
**Large-scale automated evaluation of ICU discharge summary quality reveals systematic demographic disparities**

**Confidence:** 4
**Clarity Of Writing:** good
**Clinical Significance:** good
**Methodological Novelty:** fair
**Overall Rating:** 6
**Final Rating:** 7

**Experiments And Results:**

fair

**Questions For The Authors:**

1. Can you provide validation results comparing LLM annotations with human expert assessments for a representative subset? What is the inter-rater reliability between LLM and clinical experts across the 19 metrics?
2. How stable are the prompt responses across different discharge summary lengths and clinical complexities? Did you test prompt sensitivity by varying instruction wording?
3. Have you considered correlating DS quality scores with patient outcomes such as readmission rates or care transition difficulties?
4. Did you apply multiple testing corrections given the numerous statistical comparisons?

**Strengths:**

1. The analysis of 50,000+ discharge summaries represents the largest systematic evaluation of ICU discharge summary quality to date, providing substantial statistical power and generalizability beyond previous small-scale manual reviews.
2. The adoption of established Joint Commission guidelines and structured 19-metric evaluation framework ensures clinical validity and standardization, making findings directly applicable to quality improvement initiatives.
3. The combination of numeric score-based and categorical distribution analyses strengthens statistical robustness and provides multiple perspectives on the same phenomena.
4.  Focus on ICU discharge summaries addresses a critical juncture in patient care transitions where documentation quality directly impacts patient safety and continuity of care.

**Summary Of The Paper:**

This paper presents an automated evaluation of discharge summary (DS) quality using GPT-4 to analyze 50,496 ICU discharge summaries from the MIMIC-IV dataset. The authors developed a framework based on Joint Commission guidelines, evaluating 19 metrics across five categories: Reason for Hospitalization, Significant Findings, Procedures and Treatment, Patient's Discharge Condition, and Patient and Family Instructions. Each discharge summary was annotated using a multi-step prompt engineering approach with categorical labels (Fully, Partial, Unacceptable, Missing). The paper employs both numeric score-based and categorical distribution analyses to identify quality disparities across demographic subgroups including race, insurance type, chief complaints, and admission types. Results show that while core elements like hospitalization reasons are well-documented, up to 10% of summaries lack adequate patient instructions and 3-10% have incomplete discharge condition documentation. Statistical analysis reveals significant disparities, particularly affecting Asian males with non-standard insurance coverage, where over 7% of summaries contained deficiencies compared to a 3.5% overall average.

**Weaknesses:**

1. Absence of Human Validation: The paper entirely relies on LLM annotations without providing human expert validation for this specific dataset. While citing prior work on LLM-human concordance, the lack of even subset validation significantly undermines confidence in the findings.
2. The prompt engineering section lacks reproducibility. The "final version" prompt example is incomplete, and specific instructions for the 19 metrics are not provided.
3. The paper doesn't adequately address how factors like patient complexity, length of stay, clinical urgency, or hospital unit differences might explain observed disparities.

---

### Official Review · Reviewer_YNE2 · 2025-07-19

[review text omitted: it was posted to a different submission]